# From ViT Features to Training-free Video Object Segmentation via Streaming-data Mixture Models

**Roy Uziel**
Ben-Gurion University of the Negev, Israel
uzielr@post.bgu.ac.il

**Or Dinari**
Ben-Gurion University of the Negev, Israel
dinari@post.bgu.ac.il

**Oren Freifeld**
Ben-Gurion University of the Negev, Israel
orenfr@cs.bgu.ac.il

## Abstract

In the task of semi-supervised video object segmentation, the input is the binary mask of an object in the first frame, and the desired output consists of the corresponding masks of that object in the subsequent frames. Existing leading solutions have two main drawbacks: 1) an expensive and typically-supervised training on videos; 2) a large memory footprint during inference. Here we present a training-free solution, with a low-memory footprint, that yields state-of-the-art results. The proposed method combines pre-trained deep learning-based features (trained on still images) with more classical methods for streaming-data clustering. Designed to adapt to temporal concept drifts and generalize to diverse video content without relying on annotated images or videos, the method eliminates the need for additional training or fine-tuning, ensuring fast inference and immediate applicability to new videos. Concretely, we represent an object via a dynamic ensemble of temporally- and spatially-coherent mixtures over a representation built from pre-trained ViT features and positional embeddings. A convolutional conditional random field further improves spatial coherence and helps reject outliers. We demonstrate the efficacy of the method on key benchmarks: the DAVIS-2017 and YouTube-VOS 2018 validation datasets. Moreover, by the virtue of the low-memory footprint of the compact cluster-based representation, the method scales gracefully to high-resolution ViT features. Our code is available at https://github.com/BGU-CS-VIL/Training-Free-VOS.

## 1 Introduction

Video Object Segmentation (VOS), a key computer-vision task, aims to distinguish objects of interest from the background across a sequence of video frames. In Semi-supervised VOS (SVOS), the topic of this paper, the user provides the mask of the object in the first frame (while in the case of unsupervised VOS, no mask is given), and the task is to propagate that mask to the next frames. Prior to the revolution of Deep Learning (DL), SVOS relied heavily on traditional techniques like color-based segmentation, point trajectories, and optical flow. However, the emergence of DL-based methods has significantly outpaced those traditional techniques. Thus, today DL-based methods dominate the field of SVOS. Note that according to the standard terminology in the area of VOS, the term "semi-supervised" in the name SVOS refers to the nature of the *task* (that is, the fact that a user provides the mask in the first frame), not the nature of the *solution*. In other words, solutions to the SVOS task can be either supervised, semi-supervised, or unsupervised. For example, a supervised

37th Conference on Neural Information Processing Systems (NeurIPS 2023).

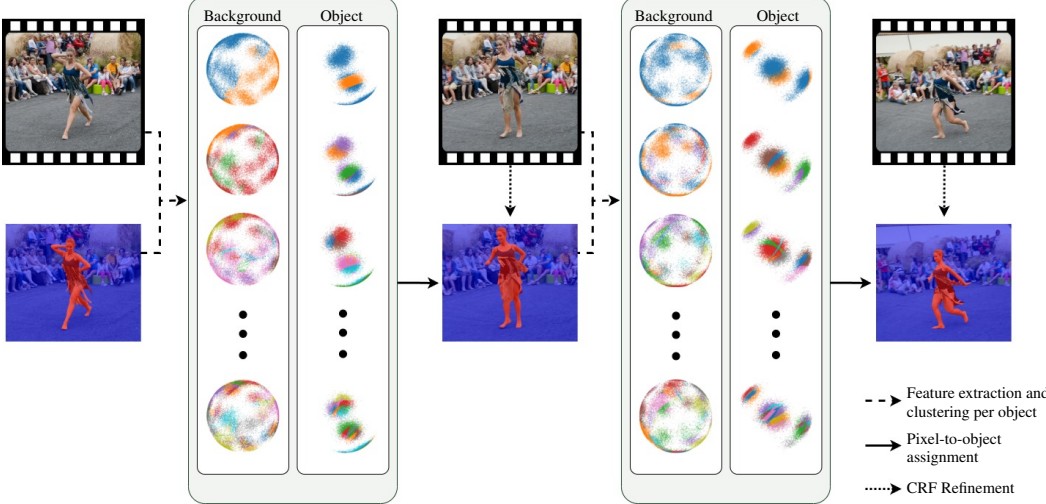

Figure 1: **Method overview.** Starting from the initial frame, the features of the object are extracted and modeled using an ensemble of von Mises-Fisher (vMF) mixtures, where each mixture has a different number of components. The higher the number of components, the finer the scale that the mixture captures. This multi-scale representation lets us capture phenomena such as drastic changes in the size/resolution of the object. The background is modeled using another such mixture ensemble. Next, all of the mixtures are updated in a streaming and memory-efficient manner at each subsequent frame. A maximum-a-posteriori rule is applied over all of the vMF components in the ensemble to generate a preliminary segmentation mask, which is then further refined by a convolutional Conditional Random Field. The iterative process progresses through the frames.

DL solution to SVOS typically requires multiple *labeled videos* to train on. In contrast, the approach proposed in this paper requires neither supervision nor learning on videos.

The main strategy employed by DL-based SVOS methods revolves around pixel-level matching to establish inter-frame correspondence. This is achieved by creating pixel-to-pixel connections between the current frame and the preceding ones. That strategy, however, often has two main limitations: 1) the need for an extensive training on large-scale (and typically annotated) video datasets; 2) a significant memory footprint due to the storage requirements of (dense representations of) previous frames for pixel matching. Addressing those prevalent challenges, we propose an unsupervised, adaptive, and compact SVOS method. By combining DL and classical techniques and capitalizing on the widespread practice of representing features as points on a hypersphere, the proposed model incorporates an ensemble of temporally- and spatially-coherent von Mises-Fisher (vMF) mixtures. The mixtures are updated dynamically in a streaming manner, together with a convolutional implementation of a Conditional Random Field (CRF).

Our method uses features from a pre-trained unsupervised deep neural network – trained on still images, not videos – but does not require training by itself as it only performs inference. The method fits multiple vMF mixtures to each object of interest (and to the background) in the initial frame. The fact that we employ multi-scale modeling (where each mixture has its own number of components) lets us represent the object at multiple levels of granularity. In subsequent frames, pixels are assigned to the most probable component (across all of the components in all of the mixtures in all of the objects). The mixture ensembles themselves are updated dynamically while taking into account temporal and spatial coherence. Following established stream clustering techniques [13], we apply a weighted window with a kernel function that reduces the weight of older frames, mitigating memory consumption and improving computational efficiency without storing extensive history of dense representations of past frames. Our model adapts to concept drifts in the objects of interest or the background. We improve spatial consistency by constraining pixel-component membership based on location and refine results using a Pixel-Adaptive Convolutional CRF (PAC-CRF) [37], fostering pixel consistency and awareness of outliers. Together with outlier-rejection mechanisms, the method mitigates error propagation across frames.

To summarize, our key contributions are: 1) We demonstrate that a synergy between classical and DL techniques can provide an effective solution for SVOS that achieves **state-of-the-art results** on the DAVIS-2017 and YouTube-VOS 2018 datasets. 2) By leveraging unsupervised pre-trained image-based features, our method eliminates the need for further training, and in particular **no training on videos is required** (unlike most competitors which require large-scale video datasets for training). 3) Our method has a **low-memory footprint** by storing only cluster-level information, as opposed to dense past-frame representations. Thus, the method does not only scale better than competing ones but also enables handling high-resolution features for higher-quality segmentation.

## 2   Related work

**The evolution of SVOS methods**. From older SVOS methods based on classic vision techniques [11, 9, 21, 50, 10, 5], the focus has shifted to DL-based SVOS methods, also known as one-shot VOS. Early semi-supervised methods focused on online fine-tuning of a pre-trained network, exemplified by OSVOS [6]. In fact, OSVOS pioneered the online fine-tuning of pre-trained networks in SVOS. However, OSVOS is prone to overfitting and fails to adapt to gradual concept drifts in the objects of interest. To address these issues, numerous extensions and refinements to OSVOS were proposed. OnAVOS [41] introduced additional fine-tuning with high-confidence frames in order to handle gradual concept drifts, albeit at the cost of increased computational complexity. Maninis *et al*. [28] integrated semantic information into the predictions via Mask R-CNN [17], thereby reducing the dependency on temporal smoothness. In an innovative departure from common practice, BubbleNets [16] proposed selecting the most suitable frame for annotation, challenging the conventional practice of annotating the first frame. Other works addressed different aspects of the OSVOS algorithm. For example, OSNM [47] utilized network modulation for more efficient fine-tuning, while A-Game [20] employed a Gaussian Mixture Model to better capture both the foreground and the background. Other noteworthy and relevant works include, but are not limited to, FRTM [35], LWL [4], and TAODA [51].

**The emergence of matching-based methods**. Recently, pixel matching methods have gained prominence due to their superior speed and results [15]. These methods, which either implicitly predict inter-frame similarities or explicitly match features between pixel pairs across frames, have been adopted by recent SVOS methods [22, 18, 3], leading to state-of-the-art performance.

**Unresolved issues and evolving developments in SVOS**. Constructing discriminative feature embeddings with temporal consistency is crucial for reliable correspondence. This is typically achieved by training backbone networks on large-scale video datasets such as OxUvA [39], YouTube-VOS [46], TrackingNet [46], and Kinetics-400 [8]. Most existing methods store dense feature representations of multiple previous frames and perform spatial correlation to propagate information across frames. However, this significantly increases memory consumption, while reducing the number of reference frames degrades the performance. Hence, finding a memory-efficient way to propagate the information is of paramount importance for achieving accurate and robust segmentation results.

## 3   Method

We start with the task definition. In the SVOS task, the user provides (only for the first frame) a binary mask for each object of interest. Let $(M_o^1)_{o=1}^{N_{\mathrm{obj}}}$ denote the masks where $N_{\mathrm{obj}}$ is the number of objects. Those (non-overlapping) $N_{\mathrm{obj}}$ masks imply another mask, $M_{N_{\mathrm{obj}}+1}^1$, which represents the background at time $t = 1$ (*i.e.*, the first frame). That mask is formed by taking the binary complement of the union of $(M_o^1)_{o=1}^{N_{\mathrm{obj}}}$. We will henceforth treat the background as just another object unless stated otherwise. For each time $t = 2, 3, \ldots, T$, where $T$ is the total number of frames, the goal is to estimate the corresponding masks, denoted by $(M_o^t)_{o=1}^{N_{\mathrm{obj}}+1}$.

Our unsupervised method for the SVOS task, depicted in Fig. 1, can be summarized as follows. We extract features from video frames (§ 3.1) and model the within-object feature distribution via a multi-scale dynamic ensemble of vMF mixtures (§ 3.2). We use a streaming-data Expectation-Maximization algorithm to update the parameters of the mixtures (§ 3.3). We assign pixels to objects based on the component-wise posterior probabilities in the entire ensemble of each object (§ 3.4). For robustness, we disregard outliers during the parameter-estimation steps, and further

improve the results by incorporating a Pixel-Adaptive Convolutional CRF (§ 3.5) for additional outlier management and segmentation refinement. For examples of qualitative results see, *e.g.*, Fig. 2.

## 3.1 Feature extraction

The feature representation we use consists of ViT features [1] and positional embeddings [40]. For the ViT features, we use the publicly-available pre-trained weights of XCiT (Cross-Covariance Image Transformer) [1], that had been trained in a self-supervised manner as outlined in [7].

Let $N_{\text{pixels}}$ be the number of pixels in each frame. Let $\widetilde{X}^t = (\widetilde{\boldsymbol{x}}_i^t)_{i=1}^N$ denote the ViT features extracted from $I_t$ (the video frame at time $t$), where $N$ is the number of features. For example, if the ViT features are extracted using non-overlapping $8 \times 8$ patches, then $N = N_{\text{pixels}}/64$. The XCiT variant, like its DINO-ViT counterpart, excels in extracting rich and discriminative features from images [2], but with an added advantage of better handling high-resolution images. In our case, we employ an XCiT model that yields a feature vector $\widetilde{\boldsymbol{x}}_i^t \in \mathbb{R}^{384}$ for each image patch $i$. To capture the spatial relationships between neighboring pixels, we extend each ViT feature, $\widetilde{\boldsymbol{x}}_i^t$, with its corresponding rotary positional embedding, $\rho(i)$, as per the methodology detailed in Su et al. [38]. This embedding represents the 2D spatial location of the ViT feature in a 64-dimensional space. Consequently, we form the extended feature vector $\boldsymbol{x}_i^t = (\widetilde{\boldsymbol{x}}i^t, w_\rho \cdot \rho(i))$ (where $w_\rho > 0$ is the weight of the positional embeddings), normalized such that $\|\boldsymbol{x}_i^t\|_{\ell_2} = 1$. We denote by $X^t = (\boldsymbol{x}_i^t)_{i=1}^N$ the collection of all these unit-length (extended) features at time $t$. Finally, given the initial masks, $(M_o^t)_{o=1}^{N_{\text{obj}}+1}$, we partition $X^1$, the features at $t = 1$, into $(X_o^1)_{o=1}^{N_{\text{obj}}+1}$ such that, for each $o \in (1, \ldots, N_{\text{obj}} + 1)$, $X_o^1$ represents the features that correspond to $M_o^1$.

## 3.2 Modeling the within-object feature distribution via a multi-scale vMF mixture

In our pursuit of capturing intricate patterns in high-dimensional data while ensuring computational efficiency, we turn to vMF mixtures; namely, a mixture model whose each component is a vMF distribution (defined below). These mixtures, known for their simple structure that depends solely on mean directions and concentration parameters, are highly efficient computationally. They are also adept at modeling data on a unit hypersphere, which is a prevalent form of representation for deep features. Since often the feature distribution of an object is not only complex but also dependent on its size (*e.g.*, when the object-to-camera distance increases, the object resolution decreases and fine details tend to disappear, which in term affects the distribution), we advocate for a multi-scale approach, using a mixture ensemble. For each object, we employ $S$ independent vMF mixture models, each corresponding to a different scale or granularity. This way, we capture details at multiple scales.

The vMF distribution is a probability distribution over $\mathbb{S}^{d-1} \triangleq \{\boldsymbol{x} : \boldsymbol{x} \in \mathbb{R}^d, \|\boldsymbol{x}\|_{\ell_2} = 1\}$, the unit sphere in $\mathbb{R}^d$. Its probability density function (pdf), evaluated at $\boldsymbol{x} \in \mathbb{S}^{d-1}$, is

$$\text{vMF}(\boldsymbol{x}; \boldsymbol{\mu}, \tau) \triangleq C_d(\tau) \exp(\tau \boldsymbol{\mu}^\top \boldsymbol{x}) \tag{1}$$

where $\boldsymbol{\mu} \in \mathbb{S}^{d-1}$ is the mean and $\tau > 0$ is the concentration parameter. The normalizer $C_d(\tau)$ depends only on $\tau$ and has a closed form that appears in **Appendix A**.

Let $s \in \{1, \ldots, S\}$. Mixture $s$ for object $o$ at time $t$ is defined as follows:

$$\sum_{k=1}^{K_s} \alpha_{o,s,k}^t \text{vMF}(x; \boldsymbol{\mu}_{o,s,k}^t, \tau_{o,s,k}^t). \tag{2}$$

Here, $\alpha_{o,s,k}^t$ represents the mixing proportions while $\boldsymbol{\mu}_{o,s,k}^t$ and $\tau_{o,s,k}^t$ are the mean and concentration parameter, respectively, of component $k$ at time $t$. The predefined number of components, $K_s$, is different for every $s$ and represents the scale of the mixture. The full set of parameters describing object $o$ at time $t$ is $\theta_o^t = \left((\alpha_{o,s,k}^t, \theta_{o,s,k}^t)_{k=1}^{K_s}\right)_{s=1}^S$, where $\theta_{o,s,k}^t = (\boldsymbol{\mu}_{o,s,k}^t, \tau_{o,s,k}^t)$.

## 3.3 Parameter estimation via Expectation-Maximization

We adapt the widely-used Expectation-Maximization (EM) framework [12] to estimate the parameters of our vMF mixtures in a streaming-data manner, tailoring it specifically for VOS. The algorithm is applied independently and in parallel to each mixture.

Our EM algorithm consists of two stages. The Expectation step (E-step) calculates the posterior probabilities of component membership (*i.e.*, feature-to-component assignment), integrating spatial coherence to improve segmentation accuracy. In the Maximization step (M-step), we update the mixture parameters to maximize the expected log-likelihood, introducing modifications to handle the streaming nature of video data and manage memory efficiently.

### 3.3.1 E-step: calculating posterior probability with spatial coherence

In the E-step we incorporate spatial coherence, a crucial factor in VOS. Let us assume that, at time $t$, pixel $i$ was assigned to object $o$ (we will later explain how pixels are assigned to objects). For each mixture $s$ of the $S$ mixtures associated with object $o$, we do the following. For every $k = 1, \ldots, K_s$, let $q_{o,s,k,i}^t$ be the posterior probability that, at time $t$, pixel $i$ belongs to component $k$ (of mixture $s$) given the observed feature $\boldsymbol{x}_i^t$ and the current estimate of $(\alpha_{o,s,k}^t, \theta_{o,s,k}^t)_{k=1}^{K_s}$. Let $N_o^t$ be the number of nonzero pixels in $M_o^t$ and, in a slight abuse of notation, let $(\boldsymbol{x}_i^t)_{i=1}^{N_o^t}$ denote the object features at time $t$. We modify the calculation of the posterior probability, $q_{o,s,k,i}^t$, by integrating a spatial constraint. This constraint is expressed as an indicator function that respects both the current parameter estimates and the spatial information inherited from the previous frame, $t - 1$. This approach ensures a smooth transition between frames and improves the spatial integrity of objects in the video sequence. Taken together, the modified posterior probability is

$$q_{o,s,k,i}^t = p(z_{o,s,i} = k | \boldsymbol{x}_i^t, \theta_{o,s,k}^t) \cdot \mathcal{I}(\boldsymbol{x}_i^t, \boldsymbol{\mu}_{o,s,k}^{t-1}) \tag{3}$$

where $z_{o,s,i}$ is the assignment of $\boldsymbol{x}_i^t$ to a component in mixture $s$ in object $o$, $p(z_{o,s,i} = k | \boldsymbol{x}_i^t, \theta_{o,s,k}^t) \propto p(\boldsymbol{x}_i^t | \theta_{o,s,k}^t) = \mathrm{vMF}(\boldsymbol{x}_i^t; \boldsymbol{\mu}_{o,s,k}^t, \tau_{o,s,k}^t)$, and the binary-valued indicator function,

$$\mathcal{I}(\boldsymbol{x}_i^t, \boldsymbol{\mu}_{o,s,k}^{t-1}) = \begin{cases} 1 & \rho(i)^\top \rho(\boldsymbol{\mu}_{o,s,k}^{t-1}) > r_s \\ 0, & \text{otherwise} \end{cases}, \tag{4}$$

returns 1 if $\boldsymbol{x}_i^t$ has a positional embedding that closely matches $\rho(\boldsymbol{\mu}_{o,s,k}^{t-1})$, the positional part of the mean of cluster $k$ from the previous frame. The quality of the match is measured in terms of cosine similarity. Note that the assignment also depends on a certain threshold, $r_s$, which in turn depends on the number of clusters, $K_s$. As $K_s$ increases, a higher degree of similarity is required, reflecting the finer granularity of the mixture. If no such pixel exists, the function returns 0, effectively preventing the current pixel from being assigned to cluster $k$. This approach ensures spatial coherence.

### 3.3.2 M-step: parameter estimation

We first provide the details for the M-step while ignoring the streaming nature of the data, and then later, in § 3.3.3, explain how we account for the streaming. In the M-step, the parameters are updated to maximize the expected log-likelihood. We iteratively update a set of sufficient statistics that encapsulate the critical properties of the data and the model. Concretely, at time $t$ we compute

$$H_{o,s,k}^t = (h_{o,s,k}^{t,1}, h_{o,s,k}^{t,2})_{k=1}^{K_s} \triangleq \left( \sum_{i=1}^{N_o^t} q_{o,s,k,i}^t, \sum_{i=1}^{N_o^t} q_{o,s,k,i}^t \boldsymbol{x}_i^t \right)_{k=1}^{K_s}. \tag{5}$$

The M-step uses well-known closed-form solutions for the estimated parameters of the vMF distribution. In particular, to estimate the concentration parameters we employ an approximation scheme based on the continued fraction method, supplemented with a correction term:

$$\boldsymbol{\mu}_{o,s,k}^t = \frac{h_{o,s,k}^{t,2}}{||h_{o,s,k}^{t,2}||}, \quad R_{o,s,k}^t = \frac{||h_{o,s,k}^{t,2}||}{h_{o,s,k}^{t,1}}, \quad \tau_{o,s,k}^t \approx \frac{d \cdot R_{o,s,k}^t - (R_{o,s,k}^t)^3}{1 - (R_{o,s,k}^t)^2}, \quad \alpha_{o,s,k}^t = \frac{h_{o,s,k}^{t,1}}{\sum_{k=1}^{K_s} h_{o,s,k}^{t,1}}. \tag{6}$$

To circumvent the computational expense of directly calculating the modified Bessel function (which appears in $C_d(\cdot)$), we adopt an alternative method to compute the log-likelihood $\log p(\boldsymbol{x}_i^t | \theta_{o,s,k}^t)$ using the mean vector $\boldsymbol{\mu}_{o,s,k}^t$ and the concentration parameter $\tau_{o,s,k}^t$. Specifically, we leverage the fact that the normalizing constant $C_d(\tau_{o,s,k}^t)$ depends solely on $\tau_{o,s,k}^t$ and can be approximated [36] by a simpler expression (see **Appendix A**). This enables us to express the log-likelihood as follows:

$$\log p(\boldsymbol{x}_i^t | \theta_{o,s,k}^t) = N_o^t \tau_{o,s,k}^t (\boldsymbol{\mu}_{o,s,k}^t)^\top \overline{\boldsymbol{x}} + N_o^t \log C_d(\tau_{o,s,k}^t), \tag{7}$$

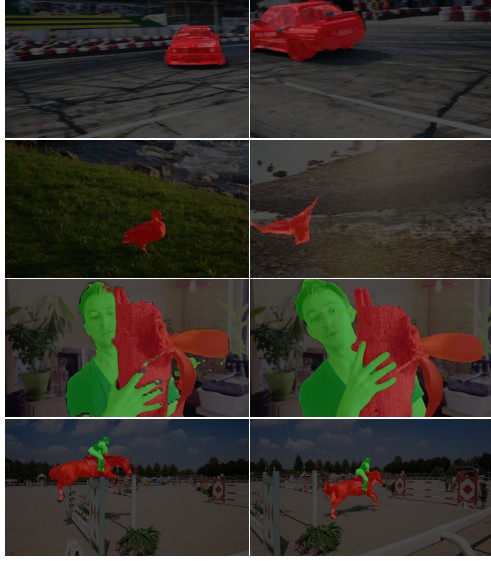

Figure 2: **Qualitative examples.**

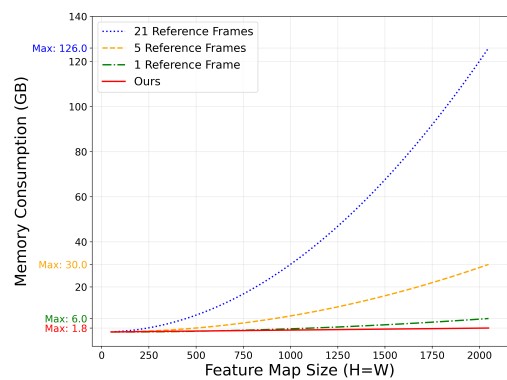

Figure 3: **Memory utilization**. This graph illustrates the memory consumption as a function of the feature map size, comparing our method with scenarios using different numbers of reference frames (1, 5, and 21). Notably, our method requires less memory than even the scenario with a single reference frame. The feature dimension for this evaluation is set to 384.

where $^\top$ denotes transpose and $\overline{\boldsymbol{x}}$ denotes the (Euclidean) sample mean of the data points. We now address the dynamic characteristics of video data.

### 3.3.3 Memory-efficient dynamic updates and temporal coherence

In the dynamic environment of video sequences, objects may experience occlusion, reemergence, or gradual appearance changes due to, *e.g.*, varying lighting conditions. Addressing these challenges, and inspired by [13], our method integrates temporal coherence into the clustering procedure using time-weighted sufficient statistics. We devise these statistics as a weighted sum of data values across multiple frames, with weights diminishing exponentially over time. By maintaining historical records of a limited length, and updating the time-weighted statistics for each frame, we efficiently manage space, memory, and time complexity. Data from the preceding frame is retained only if its (time-decaying) weight exceeds a certain threshold. Concretely, we employ a time-decaying weight $\mathcal{K}(t, t') = 2^{\lambda(t-t')}$ where the user-defined $\lambda > 0$ controls the correlation between sequential frames. As a result, older frames are assigned lower weights. Similarly to [13], we modify the sufficient statistics accordingly, and obtain a streaming-data variant of Eq. 5:

$$H_{o,s,k}^t = (h_{o,s,k}^{t,1}, h_{o,s,k}^{t,2})_{k=1}^{K_s} \triangleq \left( \sum_{t'=1}^{t} \mathcal{K}(t, t') \sum_{i=1}^{N_o^t} q_{o,s,k,i}^{t'}, \sum_{t'=1}^{t} \mathcal{K}(t, t') \sum_{i=1}^{N_o^t} q_{o,s,k,i}^{t'} \boldsymbol{x}_i^t \right)_{k=1}^{K_s}. \quad (8)$$

Our method's efficient information propagation from prior frames results in smoother transitions and improves the temporal coherence of the segmentation. Retaining (weighted) sufficient statistics from previous frames helps with the detection of reappearing, previously-occluded, objects or parts. Importantly, our memory consumption is determined by the number of clusters in the model, not $N$, the (much larger) feature map size. This drastically reduces memory usage during inference (see Fig. 3) and enables us to obtain a compact representation. Moreover, this drastic memory reduction (in comparison to methods which rely on saving history of dense representations) enables us to use higher-resolution features than the competitors (see, *e.g.*, Fig. 4). Importantly, as $\mathcal{K}(t, t')$ decreases when $|t - t'|$ increases and eventually decays to zero, in practice we store the sufficient statistics from time $t'$ only until time $t' + 15$ and then discard it.

### 3.4 Pixel-to-object assignment

Given the estimated models from time $t$, in the subsequent frame we extract features $X^{t+1}$ and assign labels to each pixel based on the mixture (from time $t$) that has the highest probability for any of its components. By employing the maximum-a-posteriori (MAP) rule, we determine the component

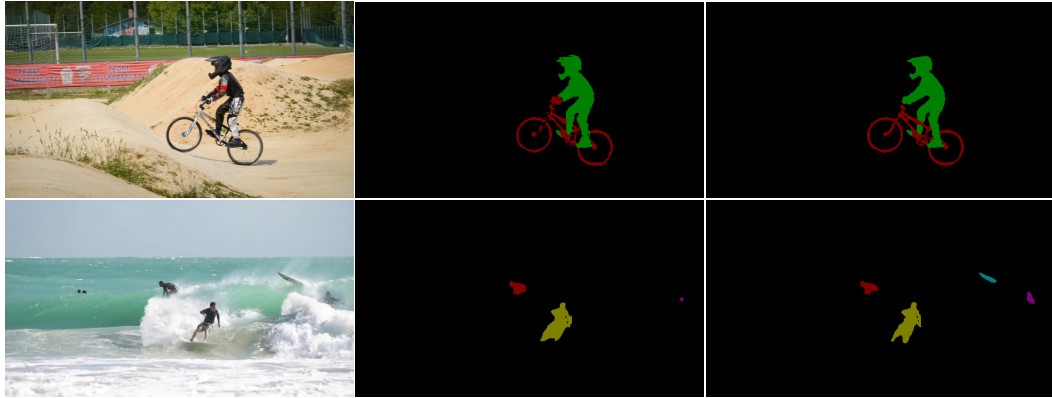

Figure 4: **Benefits of high-resolution**. Each row presents an original video frame (left) and our results obtained when using features at either low resolution (middle) or high resolution (right). Note the finer details achieved at high resolution. While it is fairly obvious that higher resolution is better, our point here is that competing methods (*e.g.*, [3, 24]) usually cannot handle such a resolution due to the increased computational and memory demands. This is unlike our method which has a low-memory footprint. Please zoom in for optimal viewing.

with the highest probability for each pixel, and then assign the pixel to the object associated with that component. In other words, for every $o \in \{1, \ldots, N_{\mathrm{obj}} + 1\}$ we compute

$$p_{i,o}^{t+1} \triangleq \Pr(\text{pixel } i \text{ at time } t+1 \text{ belongs to object } o) \propto \max_{s:s\in 1,\ldots,S} \left( \max_{k:k\in 1,\ldots,K_s} p(\boldsymbol{x}_i^{t+1}|\theta_{o,s,k}^t) \right) \quad (9)$$

and then assign the pixel to object according to $\arg\max_{o:o\in\{1,\ldots,N_{\mathrm{obj}}+1\}} p_{i,o}^{t+1}$. Next, we generate the (non-overlapping) masks for time $t+1$, $(M^{t+1})_{o=1}^{N_{\mathrm{obj}}+1}$ via

$$M_o^{t+1}(i) = \begin{cases} 1, & \text{if } o = \arg\max_{o'} p_{i,o}^{t+1} \\ 0, & \text{otherwise} \end{cases} \quad (10)$$

where $M_o^{t+1}(i)$ denotes pixel $i$ of $M_{o'}^{t+1}$.

### 3.5 Refining the segmentation using outlier rejection and a convolutional CRF

Recall that the initial mask is the sole ground truth reference that the method has access to. Thus, as the video sequence unfolds, the presence of outliers poses a substantial challenge to the accuracy of the clustering process. These outliers, which stem from factors such as occlusions, illumination changes, or noise, might hurt the integrity of existing clusters or lead to confusing them with unrelated clusters. Our approach for mitigating such potential sources of error incorporates two strategies: outlier rejection and a refinement using a convolutional CRF.

In the initial phase, we exclude points from the parameter-estimation step when the pixel-to-object assignment is inconclusive. Specifically, if the score of a pixel's second-best match closely rivals that of its best match, we consider that pixel an outlier. We then employ the Pixel-Adaptive Convolutional CRF (PAC-CRF) [37] to further refine the segmentation. PAC-CRF, with its fixed window connections around each pixel, effectively filters noise while preserving segmentation coherence. It bolsters our outlier handling strategy by enabling the exclusion of points where the PAC-CRF prediction conflicts with our initial prediction. After several EM steps (where the M step excludes the outliers), we apply PAC-CRF again, leveraging the source image to improve the pixel-wise accuracy and the segmentation quality. Furthermore, we leverage the learnability of the PAC-CRF's pairwise potentials' weights and optimize them using Focal-Loss [26, 30]. This optimization, carried out over a few gradient steps with the ground-truth annotation from the first frame, ensures that our method adapts well to the given data, leading to improved performance and more accurate segmentation maps (Fig. 2).

## 4 Results

In this section, we present the performance of the proposed method on the DAVIS-2017 and YouTube-VOS datasets. A subsequent subsection delves into our adeptness at harnessing features from deeper

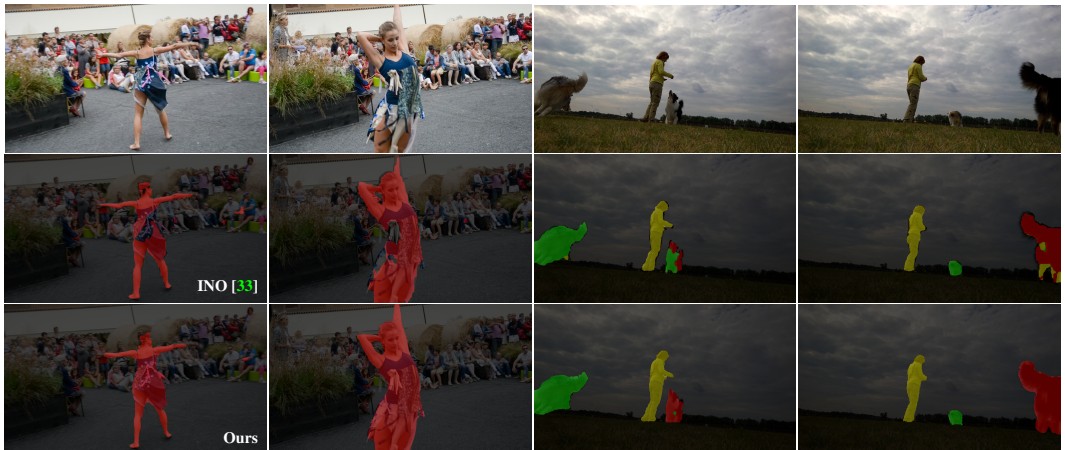

Figure 5: **Qualitative examples on DAVIS-2017**, comparing our results (third row) with one of the key competitors (second row).

Table 1: **Results** on DAVIS-2017 validation in terms of the mask ($\mathcal{J}$) and boundary ($\mathcal{F}$) accuracy (IoU). The subscript $[\cdot]_r$ denotes the recall of the metric, while $[\cdot]_m$ signifies the mean.

| Method | Training Video Dataset (Duration) | $\mathcal{J}\&\mathcal{F}_m$ | $\mathcal{J}_m$ | $\mathcal{J}_r$ | $\mathcal{F}_m$ | $\mathcal{F}_r$ |
|---|---|---|---|---|---|---|
| *Unsupervised* | | | | | | |
| Colorization[42] | Kinetics (800 hours) | 34.0 | 34.6 | 34.1 | 32.7 | 26.8 |
| CorrFlow[23] | OxUvA (14 hours) | 50.3 | 48.4 | 53.2 | 52.2 | 56.0 |
| TimeCycle[44] | VLOG (344 hours) | 48.7 | 46.4 | 50.0 | 50.0 | 48.0 |
| UVC[25] | Kinetics (800 hours) | 60.9 | 59.3 | 68.8 | 62.7 | 70.9 |
| MuG[27] | OxUvA (14 hours) | 54.3 | 52.6 | 57.4 | 56.1 | 58.1 |
| MAST[22] | Youtube-VOS (5.58 hours) | 65.5 | 63.3 | 73.2 | 67.6 | 77.7 |
| CRW[18] | Kinetics (800 hours) | 68.3 | 65.5 | 78.6 | 71.0 | 82.9 |
| ContrastCorr[43] | TrackingNet (140 hours) | 63.0 | 60.5 | - | 65.5 | - |
| VFS[45] | Kinetics (800 hours) | 66.7 | 64.0 | - | 69.4 | - |
| JSTG[49] | Kinetics (800 hours) | 68.7 | 65.8 | 77.7 | 71.6 | 84.3 |
| DULVS[3] | TrackingNet (140 hours) | 69.4 | 67.1 | 80.9 | 71.7 | 84.8 |
| CLTC[19] | Youtube-VOS (5.58 hours) | 70.3 | 67.9 | 78.2 | 72.6 | 83.7 |
| DINO[7] | **No training video is required** | 71.4 | 67.9 | 80.7 | 74.9 | 87.8 |
| LIIR[24] | Youtube-VOS (5.58 hours) | 72.1 | 69.7 | 81.4 | 74.5 | 85.9 |
| INO[33] | Kinetics (833 hours) | 72.5 | 68.7 | 82.0 | 76.3 | 89.0 |
| Ours | **No training video is required** | **76.3** | **73.8** | **85.6** | **78.7** | **89.2** |
| *Supervised* | | | | | | |
| OSVOS[6] | DAVIS-2016 (0.5 hours) | 60.3 | 56.6 | 63.8 | 63.9 | 73.8 |
| TVOS[48] | DAVIS-2017 + Youtube-VOS (6 hours) | 72.6 | 69.9 | - | 74.7 | - |
| STM[31] | DAVIS-2017 + Youtube-VOS (6 hours) | 81.7 | 79.2 | - | 84.3 | - |

networks and superior resolutions. Video demonstrations can be found here. Details regarding our implementation are available in **Appendix B**.

### 4.1 Evaluation on DAVIS-2017 and YouTube-VOS

We report the results on two widely-used SVOS benchmarks: YouTube-VOS [46] and DAVIS 2017 [34]. Commonly used metrics for assessing VOS performance include the Jaccard's index (J) [14], which calculates the intersection-over-union (IoU) of the object mask, and the F-measure (F) [29], which assesses the contour accuracy. These metrics provide comprehensive insight into the segmentation performance by evaluating both the object area and contour precision. We report the mean IoU for mask and contour, denoted as $\mathcal{J}_m$ and $\mathcal{F}_m$, as well as the recall $\mathcal{J}_r$ and $\mathcal{F}_r$, which are

| Table 2: **DAVIS-2017 (train)** | | | | | |
|---|---|---|---|---|---|
| Method | $\mathcal{J}\&\mathcal{F}_m$ | $\mathcal{J}_m$ | $\mathcal{J}_r$ | $\mathcal{F}_m$ | $\mathcal{F}_r$ |
| DULVS | 68.0 | 64.4 | 75.4 | 71.7 | 81.3 |
| LIIR | 63.9 | 60.3 | 69.7 | 67.5 | 75.4 |
| INO | 67.7 | 62.4 | 71.7 | 73.0 | 84.1 |
| Ours | **73.9** | **80.0** | **77.9** | **77.9** | **88.1** |

| Table 3: **DAVIS-2017 (test-dev)** | | | | | |
|---|---|---|---|---|---|
| Method | $\mathcal{J}\&\mathcal{F}_m$ | $\mathcal{J}_m$ | $\mathcal{J}_r$ | $\mathcal{F}_m$ | $\mathcal{F}_r$ |
| DULVS | 58.3 | 54.5 | 64.7 | 62.0 | 70.1 |
| LIIR | 50.3 | 47.6 | 54.7 | 52.9 | 60.2 |
| INO | 61.9 | 56.5 | 63.8 | 67.3 | 77.6 |
| Ours | **66.5** | **63.3** | **74.5** | **69.6** | **80.1** |

calculated using an IoU threshold of 0.5. The metric $\mathcal{J}\&\mathcal{F}_m$ represents the average of $\mathcal{J}_m$ and $\mathcal{F}_m$. For a fair comparison, we use the same video resolution as the competing methods, despite the fact that our method can process higher resolutions than them.

**DAVIS 2017**. The DAVIS-2017 dataset [34], featuring high-quality videos with diverse object appearances and challenging motion patterns, is a widely-accepted SVOS benchmark. On the DAVIS-2017 validation set (Table 1), our method demonstrated top-tier scores in the Jaccard index, F-measure, and $\mathcal{J}\&\mathcal{F}_m$ metrics. Specifically, our results include a $\mathcal{J}\&\mathcal{F}_m$ score of 76.3, $\mathcal{J}_m$ of 73.8, $\mathcal{J}_r$ of 85.6, $\mathcal{F}_m$ of 78.7, and $\mathcal{F}_r$ of 89.2. These results clearly outperform the other methods (see also Fig. 5). We also evaluated our method on the DAVIS-2017 training and test-dev sets, collectively constituting 90 additional sequences. These sets allowed for a broader performance assessment, with the training set offering an extra dimension of evaluation since it is not included in the training data of the methods we compared with. Results from the test-dev set, officially validated by the evaluation server, further demonstrate our method's efficacy (Table 2 & Table 3). Qualitative results from these extended evaluations are available in **Appendix C**.

**YouTube-VOS**. The YouTube-VOS dataset [46] serves as a large-scale benchmark for SVOS, introducing a variety of complexities such as occlusions, scale changes, and motion blur. Unique to this dataset is the division of object classes between the training and validation sets, where only a subset is shared, allowing the benchmark to distinguish between 'seen' and 'unseen' categories. Our method's scores, obtained through the official evaluation server, showcase its adaptability in navigating these complex situations, surpassing the performance of several state-of-the-art approaches (Table 6).

## 4.2 Performance and feature analysis

Our method adeptly processes high-resolution features, distinguishing it from other models constrained by memory, as illustrated in Table 4. While merely adopting deeper backbones does not guarantee superior outcomes [45], our model showcases its capability to harness the strengths of DINOv2's [32] features. To counteract the reduced spatial information inherent in DINOv2 due to its larger patch size, We concatenated the DINOv2 features with the XCiT features we had previously used. This approach ensures a rich and comprehensive representation of the visual scene. As demonstrated in Table 5, our method effectively capitalizes on the enhanced features from both deeper backbones and higher spatial resolutions.

Table 4: **FPS across resolutions.** Comparison on Tesla V100-32GB, excluding feature extraction. OOM stands for "Out Of Memory".

| Resolution | Feat. Size | Ours | [33] | [3] |
|---|---|---|---|---|
| | 384 | **8.25** | 0.13 | 0.14 |
| $120 \times 210$ | 768 | **6.50** | 0.11 | 0.12 |
| | 1152 | **5.25** | 0.10 | 0.11 |
| | 384 | **1.55** | OOM | OOM |
| $240 \times 420$ | 768 | **1.28** | OOM | OOM |
| | 1152 | **1.10** | OOM | OOM |

Table 5: **DAVIS-2017 (val) with DINOv2 features.** Fusion with our features elevates performance. "$\times 2$" denotes using features with twice the spatial resolution. DINOv2 "S" and "B" stand for "small" and "base", respectively.

| Method | $\mathcal{J}\&\mathcal{F}_m$ | $\mathcal{J}_m$ | $\mathcal{F}_m$ |
|---|---|---|---|
| + DINO2-S | 0.776 | 0.754 | 0.798 |
| + DINO2-B | 0.782 | 0.760 | 0.804 |
| $\times 2$ + DINO2-S | 0.784 | 0.762 | 0.806 |
| $\times 2$ + DINO2-B | **0.803** | **0.780** | **0.825** |

## 4.3 Ablation study

We performed an ablation study (Fig. 6) to analyze the influence of different parts of the method on the performance. The combination of outlier detection and segmentation refinement showed a

Table 6: **Results on YouTube-VOS 2018 (val)**

| Method | Seen | | Unseen | | Mean |
|---|---|---|---|---|---|
| | $\mathcal{J}_m$ | $\mathcal{F}_m$ | $\mathcal{J}_m$ | $\mathcal{F}_m$ | |
| *Unsupervised* | | | | | |
| Colorize[42] | 43.1 | 38.6 | 36.6 | 37.4 | 38.9 |
| CorrFlow[23] | 50.6 | 46.6 | 43.8 | 45.6 | 46.6 |
| MAST[22] | 63.9 | 64.9 | 60.3 | 67.7 | 64.2 |
| DULVS[3] | 70.2 | 71.9 | **66.5** | 74.8 | 70.6 |
| CLTC[19] | 66.2 | 67.9 | 63.2 | 71.7 | 67.3 |
| LIIR[24] | 67.9 | 69.7 | 65.7 | 73.8 | 69.3 |
| INO[33] | 70.7 | 73.2 | 65.6 | **75.6** | 71.3 |
| Ours | **72.0** | **74.2** | 66.2 | 73.7 | **71.5** |
| *Supervised* | | | | | |
| OSVOS[6] | 59.8 | 60.5 | 54.2 | 60.7 | 58.8 |
| TVOS[48] | 67.1 | 69.4 | 56.5 | 63.0 | 71.6 |
| STM[31] | **79.7** | **84.2** | **72.8** | **80.9** | **79.4** |

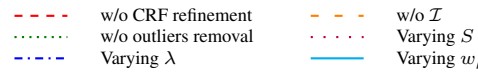

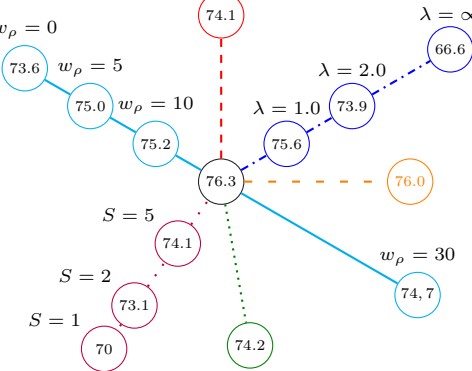

Figure 6: **Ablation study of model parameters.** We report the $\mathcal{J}\&\mathcal{F}_m$ score evaluated on DAVIS-2017 val. Our baseline configuration (the centered one) is: $S = 10$, $\lambda = 0.33$, $w_\rho = 15$.

synergistic improvement in performance, with each contributing significantly when implemented separately. The necessity of our streaming mechanism was underscored when we observed a notable performance drop while solely relying on the previous frame for initializing mixture parameters ($\lambda = \infty$). Furthermore, examining the number of mixtures per object, we notice a substantial performance boost when increasing the number of mixtures from one to two, underscoring the importance of multi-scale object representations. However, the added benefit started to decay with further increase, leading us to settle on $S = 10$ mixtures in our reported results. Concerning spatial components, removing positional embeddings (*i.e.*, $w_\rho = 0$) resulted in a major performance drop, confirming their essential role. Additionally, the introduction of the indicator function led to an additional positive effect on performance.

## 5 Conclusion

We proposed a novel SVOS approach that achieves state-of-the-art results on challenging benchmarks. Our memory-conscious strategy also opens possibilities for processing high-resolution videos, suggesting a valuable avenue for future video understanding research.

### 5.1 Limitations

The proposed method faces difficulties in two scenarios (see **Appendix D** for examples). The first is related to re-identification challenges, where occlusion between multiple similar instances of the same object type (*e.g.*, two camels) makes it challenging to assign correct labels to each instance. This limitation stems from the method's lack of explicit object identity representation and modeling of temporal dynamics, which could help disambiguate overlapping objects. The second scenario occurs in the presence of a strong motion blur. The method is sensitive to such a blur, likely because the ViT features were trained on still images, not videos. This limitation becomes more pronounced when motion blur persists across consecutive frames, as then the error propagation is more significant.

## Acknowledgments and disclosure of funding

This work was supported by: the BGU CS Lynn and William Frankel Center; the Israeli Council for Higher Education via the BGU Data Science Research Center; Israel Science Foundation Personal Grant #360/21.

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
