# OpenReview forum: "From ViT Features to Training-free Video Object Segmentation via Streaming-data Mixture Models"
_NeurIPS.cc/2023/Conference — NeurIPS 2023 poster_

### Official Review · Reviewer_XSWn · 2023-06-27

**Soundness:** 3 good
**Presentation:** 3 good
**Contribution:** 3 good
**Rating:** 6
**Confidence:** 4

**Summary:**

This paper proposes a novel method for semi-supervised video object segmentation. The method combines pre-trained deep features from still images with streaming-data clustering techniques to model the object and the background as dynamic ensembles of von Mises-Fisher mixtures. The method does not require any additional training or fine-tuning on videos, and has a low memory footprint by storing only cluster-level information. The method also incorporates spatial coherence, outlier rejection, and convolutional conditional random fields to improve the segmentation quality. The paper demonstrates that the method achieves state-of-the-art results on two challenging benchmarks: DAVIS-2017 and YouTube-VOS 2018.




**Strengths:**

The paper shows that using pre-trained features from still images can eliminate the need for costly and supervised training on videos, while using streaming-data clustering can adapt to temporal changes and reduce memory consumption.
1. The idea of using vMF distribution to model the changes of features in stream data is interesting.
2. The memory-conscious strategy opens possibilities for processing long videos, while the current solution is limited to videos with only about one hundred frames.

**Weaknesses:**

1. Almost all of the methods compared in this work target correspondence learning, which can not only handle VOS but also pose/object tracking. Thus it is reasonable that this work specifically tailored for VOS will achieve SOTA performance.
2. Though no training video is required, the backbone has to be pre-trained on million static images which is much larger than and has more diverse scenes than YouTube-VOS or Kinetics.
3. The improvement of performance on YouTube-VOS 2018 is not as considerable as that on DAVIS17 val. Moreover, it is even inferior to [1] on YouTube-VOS 2018 (i.e., 71.5 v.s. 72.4) with a stronger backbone (i.e., XCiT-small v.s. ResNet-50).
4. How about the inference speed of the proposed method compared to existing work? Will multiple vMF mixture models impose a tremendous burden on inference?
5. Please check the bib carefully, e.g. [7] and [8] are the same.

[1]. Unified Mask Embedding and Correspondence Learning for Self-Supervised Video Segmentation. CVPR23

**Questions:**

Please refer to weakness.

**Limitations:**

The limitation has been well discussed in Supplemental Material.

---

> ### Author Rebuttal · Authors · 2023-08-09
>
> We thank the reviewer for the thoughtful, useful, and positive review.
>
> - Our claim is for SOTA results only on the VOS task, and we make no claims to solve other tasks. However, the methods we cited and compared to are the current SOTA (which we beat…) in VOS, so we had to compare with them even if they can also solve non-VOS tasks.
> - The uniqueness of our approach stems from its capability to understand video dynamics without any exposure to video sequences during training. This is a non-trivial task as it necessitates the model to contend with video-specific attributes such as occlusion, camera movement, and changes in object appearances, which are absent in static images. Therefore, the proficiency to learn from static images and apply this knowledge effectively to video sequences enhances the novelty and complexity of our method.
> - The evaluation of our method on the YouTube-VOS 2018 dataset yielded a score of 71.5, slightly below [1]'s score of 72.4. It's significant to note that [1] utilized the same self-supervised pretraining scheme as [2], followed by additional specialized training on this dataset, whereas the pre-trained features used by our method were trained using only static images. However, this specialized approach by [1] didn't result in superior performance on the DAVIS2017 validation and test sets. Moreover, [1] demonstrates limitations in frame rate and an inability to utilize higher-resolution features, as discernible from their pseudo code. In contrast, our method's performance underscores its more robust adaptability across different datasets and superior handling of high-resolution features, even when solely employing static images during training.
> Furthermore, when we examine the performance of our method against INO, which employs a backbone of similar strength to ours (ViT B/8), our method still delivers superior results. This direct comparison demonstrates that our results aren't solely a product of the powerful backbone.
> - Performance Comparison in Frames Per Second (FPS):
>
>   | Resolution       | Feature Dimension | Ours  | INO  | DULVS |
>   |------------------|-------------------|-------|------|-------|
>   | 1/4 (120 x 210)  | 384               | 8.25  | 0.13 | 0.14  |
>   |                  | 768               | 6.50  | 0.11 | 0.12  |
>   |                  | 1152              | 5.25  | 0.10 | 0.11  |
>   | 1/2 (240 x 420)  | 384               | 1.55  | OOM  | OOM   |
>   |                  | 768               | 1.28  | OOM  | OOM   |
>   |                  | 1152              | 1.10  | OOM  | OOM   |
>
>   _OOM: Out of Memory_
>
>   We evaluated three models using feature maps at resolutions of 1/4 and 1/2 of the image size (480 x 840) across three feature dimensions (384, 768, 1152). This comparison involved computing the frames per second (FPS) after the first 21 frames, exclusive of the feature extraction step. Our model consistently outperformed the others, exhibiting particular strength in higher resolutions where competing models faced out-of-memory (OOM) issues.
>   (Tested using Tesla V100-32GB)
>
>
> - [1] Li, Liulei, et al. "Unified Mask Embedding and Correspondence Learning for Self-Supervised Video Segmentation." Proceedings of the IEEE/CVF Conference on Computer Vision and Pattern Recognition. 2023.
> - [2] Caron, Mathilde, et al. "Emerging properties in self-supervised vision transformers." Proceedings of the IEEE/CVF international conference on computer vision. 2021.

---

> > ### Comment · Reviewer_XSWn · 2023-08-14
> >
> > Thanks for your response, all of my concerns have been addressed.

---

### Official Review · Reviewer_He22 · 2023-07-02

**Soundness:** 3 good
**Presentation:** 3 good
**Contribution:** 3 good
**Rating:** 7
**Confidence:** 3

**Summary:**

This paper tackles the semi-supervised video object segmentation problem. It presents a method that relies on clustering features from a pre-trained ViT model. The presented method has a low memory footprint and does not need any additional training. It shows SOTA results on DAVIS-2017 and YouTube-VOS 2018 validation datasets.

**Strengths:**

The presented algorithm is the first to show a low memory footprint and indeed can scale easily.

The paper presents a comprehensive study of the different components of the algorithm and an interesting ablation that shows the necessity of the different components.

The results of the paper are both visually and numerically impressive.

**Weaknesses:**

While the algorithm in the paper results in impressive results, it is hard to follow all the notations and the optimization objectives, mostly due to the nested indexing. While I was able to follow it eventually, I suggest rewriting it (one possibility is to start with a simple 2 classes foreground and background case -  and extend it later to N classes).

One element that is not ablated in this paper, is the quality of the pre-trained model and the features that are used. To show that this algorithm can improve and produce better results in the future, one should show how the performance of SVOS improves when "better" features are given to it. e.g. does a ViT model that was pre-trained on more data result in better downstream performance? Does a larger model improve the SVOS? Does the algorithm improve with higher resolution features? All of these questions are important for understanding if this algorithm will survive the "test of time".

**Questions:**

Most of my questions are mentioned above.
Another question that I will like to know the answer to is about the time consideration - how much time does it take, as opposed to other methods for SVOS, and how does it scale with the spatial size of the features?

**Limitations:**

The limitations of the paper are discussed and strongly relate to the choice of the pre-trained model and its corresponding features, and might be addressed by using a model pre-trained on videos.

---

> ### Author Rebuttal · Authors · 2023-08-09
>
> We thank the reviewer for the thoughtful, useful, and positive review.
>
> **Adopting Deeper Networks:**
> It is important to note that simply adopting deeper networks doesn't necessarily guarantee superior results. As observed in previous self-supervised learning approaches like VFS [1], performance sometimes remains unchanged or even degrades when using deeper backbones like ResNet-50. It can even introduce additional complications which already exist during training, such as issues with memory and convergence. Thus, our method provides an effective and much-needed solution to a practical problem.
>
> The table below represents an evaluation conducted on the DAVIS-2017 validation set.
>
> | Features                | J&F-Mean | J-Mean  | J-Recall | F-Mean  | F-Recall |
> |-------------------------|----------|---------|----------|---------|----------|
> | ours+dinov2_small       | 0.776    | 0.754    | 0.854    | 0.798    | 0.890    |
> | ours+dinov2_base        | 0.782    | 0.760    | 0.880    | 0.804    | 0.900    |
> | oursx2+dinov2_small     | 0.784    | 0.762    | 0.873    | 0.806    | 0.905    |
> | oursx2+dinov2_base      | 0.803    | 0.780    | 0.885    | 0.825    | 0.916    |
>
> In the experiment, DINOv2's [2] features were concatenated with our original features, a necessity due to DINOv2's large patch size of 14. This fusion provided more effective information utilization. The results showed a marked improvement in J&F-Mean with the "oursx2" configurations (features with higher spatial dimensions). This improvement is attributed to our method's ability to exploit and use higher resolutions, facilitated by XCiT's capability to handle them. A comparison between the DINOv2 versions ("small" vs "base") revealed that the "base" version consistently performed better, underscoring the influence of stronger pre-trained models. Overall, these findings demonstrate that the method's performance can be elevated by using higher resolutions and selecting more advanced pre-trained models, delineating a promising direction for future enhancements.
>
> - [1] Xu, Jiarui, and Xiaolong Wang. "Rethinking self-supervised correspondence learning: A video frame-level similarity perspective." Proceedings of the IEEE/CVF International Conference on Computer Vision. 2021.
> - [2] Oquab, Maxime, et al. "Dinov2: Learning robust visual features without supervision." arXiv preprint arXiv:2304.07193 (2023).

---

> > ### Comment · Reviewer_He22 · 2023-08-20
> >
> > Thanks for your response, all of my concerns have been addressed.

---

> ### Comment · Area_Chair_BjJk · 2023-08-20
> **Reminder from AC**
>
> Dear Reviewer
>
> Could you please check the rebuttal, if you have further concerns ?
>
> Best,
> AC

---

### Official Review · Reviewer_sD4x · 2023-07-06

**Soundness:** 3 good
**Presentation:** 3 good
**Contribution:** 3 good
**Rating:** 5
**Confidence:** 5

**Summary:**

This paper proposes a training free framework for semi-supervised VOS. Here, the features of objects from a pretrained ViT are represented as vMF, where the objects across frames are associated to perform propagation from initial masks. The results are better than previous self-supervised methods that trained on unlabeled videos.

**Strengths:**

The idea of associating objects in vMF space for SVOS is interesting, which is a new insight different from the previous contrastive learning based self-supervised methods. The paper is well organized and sufficient comparison visual results are given.

**Weaknesses:**

-The claimed limitations in SVOS is not true especially for the large memory footprint. Actually, some lightweight designs have been proven in VOS like mobilevos[1*] and AOT[2*]. Here, [1*] focuses a lightweight for real-time vos and [2*] tackles the memory storage issue in STM and the repeated inference for each object id at testing. It's true that the heuristic association method gets rid of the usage of complicated decoders where temporal correspondence and mask generation are performed. However, the large memory is not only from the network parameters but the computational costs. For now, the authors did't give convincing results to demonstrate this contribution like comparisons in terms of params, gflops or fps.
-For the temporal coherent, is it possible to extend to model both inter-object contrast and intra-object consistency, so the method can directly predict multiple instances at once.
--Although the method is training free, many hyperparameters need to be tuned during testing. Figure 5 also demonstrates that the change of hyperparameters would impact the final performance.

[1*] MobileVOS: Real-Time Video Object Segmentation Contrastive Learning meets Knowledge Distillation, cvpr23.
[2*] Associating Objects with Transformers for Video Object Segmentation, neurips 21.

**Questions:**

-More details should be illustrated in Table 1 and Table 2, e.g., backbone, resolutions. The proposed method uses ViT while the others always use res18 or res50. It is necessary to demonstrate that the superior results come from the well designed algorithm instead of the usage of more powerful backbone.


**Limitations:**

The authors discussed the limitations of their methods.

---

> ### Author Rebuttal · Authors · 2023-08-09
>
> We thank the reviewer for the thoughtful, useful, and overall-positive review.
>
> - **Large memory footprint**. MobileVOS and AOT are supervised methods that indeed accomplish admirable memory management and computational efficiency. However, these achievements are grounded in the utilization of training labels, which streamline label propagation, subsequently reducing computational and memory demands. In contrast, our approach operates within the unsupervised realm, where label propagation is inherently more complex and memory-intensive.
> It's also noteworthy that, to our knowledge, no existing unsupervised methods have effectively navigated the memory-intensive challenges associated with label propagation. Our method bridges this gap, providing an innovative solution in the unsupervised paradigm.
>
>   **Performance Comparison in Frames Per Second (FPS):**
>
>   | Resolution       | Feature Dimension | Ours  | INO  | DULVS |
>   |------------------|-------------------|-------|------|-------|
>   | 1/4 (120 x 210)  | 384               | 8.25  | 0.13 | 0.14  |
>   |                  | 768               | 6.50  | 0.11 | 0.12  |
>   |                  | 1152              | 5.25  | 0.10 | 0.11  |
>   | 1/2 (240 x 420)  | 384               | 1.55  | OOM  | OOM   |
>   |                  | 768               | 1.28  | OOM  | OOM   |
>   |                  | 1152              | 1.10  | OOM  | OOM   |
>
>   _OOM: Out of Memory_
>
>   We evaluated three models using feature maps at resolutions of 1/4 and 1/2 of the image size (480 x 840) across three feature dimensions (384, 768, 1152). This comparison involved computing the frames per second (FPS) after the first 21 frames, exclusive of the feature extraction step. Our model consistently outperformed the others, exhibiting particular strength in higher resolutions where competing models faced out-of-memory (OOM) issues.
>   (Tested using Tesla V100-32GB)
> - **Increasing the inter-object variability and increasing the intra-object similarity**. Please note that our method does not make any assumption about the object class. Thus, it is not designed for instance segmentation per se. That said, and as demonstrated in many of our videos (please also visit the url provided with the paper which shows many such videos), if the user provides multiple masks in the first frame, then our method successfully segments multiple objects, including different instances of the same class (e.g. see the video with several dogs).
> - **Hyperparameters** exist in most machine-learning models. The number of hyperparameters in our method is not higher than, e.g., in a typical pure deep-learning method. In comparison to traditional training-based models, tuning hyperparameters in our approach is in fact decidedly more straightforward. Of note, we used the same values of hyperparamters for both Youtube-VOS and Davis, even though these datasets differ from each other in multiple ways.
> - **Backbones and Resolutions:**
>
>   | Model  | Backbone   | Resolution          |
>   |--------|------------|---------------------|
>   | DINO   | ViT B/8    | ⅛ of image resolution |
>   | LIIR   | Resnet-18  | ¼ of image resolution |
>   | VFS    | Resnet-50  | ⅛ of image resolution |
>   | DULVS  | Resnet-18  | ⅛ of image resolution |
>   | INO    | ViT B/8    | ⅛ of image resolution |
>
>   In the revised version of our paper, we will ensure these details are included.
> - **Adopting Deeper Networks:**
> It is important to note that simply adopting deeper networks doesn't necessarily guarantee superior results. As observed in previous self-supervised learning approaches like VFS [1], performance sometimes remains unchanged or even degrades when using deeper backbones like ResNet-50. It can even introduce additional complications which already exist during training, such as issues with memory and convergence. Thus, our method provides an effective and much-needed solution to a practical problem.
> To further dispel any concerns about the effectiveness of our algorithm, we direct  attention to our comparison with INO, which employs ViT B/8. Despite using the same powerful backbone, our method outperforms INO, thereby demonstrating that our superior results are indeed a product of our well-designed algorithm, and not merely the result of utilizing a more powerful backbone.
>
> - [1] Xu, Jiarui, and Xiaolong Wang. "Rethinking self-supervised correspondence learning: A video frame-level similarity perspective." Proceedings of the IEEE/CVF International Conference on Computer Vision. 2021.

---

> > ### Comment · Reviewer_sD4x · 2023-08-14
> >
> > Thanks for your response. I don't have more questions.

---

### Official Review · Reviewer_RHPb · 2023-07-09

**Soundness:** 4 excellent
**Presentation:** 4 excellent
**Contribution:** 3 good
**Rating:** 5
**Confidence:** 3

**Summary:**

The method tries to combine classical technique for SVOS task namely vMF and CRFs with advance Deep Learning based ViT representations. By doing so, the proposed method eliminates the training requirements on video data. The authors perform comprehensive experimentation to evaluate their approach.

**Strengths:**

The paper is well written and easy to follow. Authors have done a thorough job in providing comprehensive experimental evaluation for their technique. The major strength of this approach is it does not require video training data.

**Weaknesses:**

- The performance of the model decreases when it encounters the unseen examples as can be observed from the table 2. If the proposed model does not require training then why would the performance on unseen examples is lower?
-  I'm not sure I understand what Fig. 3 represents? Is it a comparison of memory utilization by proposed method and baselines that require 1, 5, 21 reference frames?
- The claim of low-memory footprint is unclear to me? is the claim made with respect to other baselines? If so, by how much delta is the proposed method better?

**Questions:**

- As claimed by authors, their method has a low-memory footprint. How does it compare to the baseline in regard to the speed of performing the VOS over a video? I want to know if there is a trade-off between the memory footprint and performance speed.

**Limitations:**

As the authors are using pre-trained VITs to extract the features. All the limitations of VITs carry forward to this approach.

---

> ### Author Rebuttal · Authors · 2023-08-09
>
> We thank the reviewer for the thoughtful, useful, and overall-positive review.
>
> -  **_"The performance of the model decreases when it encounters unseen examples"_**.
> The decrease in performance on unseen examples, as noted in Table 2, does not stem from a need for training, but rather the complexity of these examples. The unseen categories likely comprise more challenging cases. This observation aligns with other unsupervised models not trained on the YouTube-VOS 2018 dataset, where a similar drop in performance occurs. This pattern emphasizes that the challenge lies in the unique difficulty of the unseen examples, rather than a specific limitation of our model.
>
> - **Clarification on Figure 3 and the memory footprint**.
> Figure 3 compares memory consumption for the retention of history information between our proposed method and recent works in a similar setting (unsupervised, correspondence based matching), which often need up to 21 reference frames.
> The cumulative memory utilization during inference is affected by:
>     1.  **Memory Storage**: Traditional methods store multiple dense feature maps from prior frames, increasing the memory footprint based on the quantity and size of these maps. Our method significantly reduces this by tying memory usage to the number of clusters, which demands far less memory than even a single dense feature map.
>     2. **Computational Load**: Standard label propagation techniques compute cosine similarity for each feature point in the current frame with all feature points within a certain radius in all reference frames. That approach struggles with scalability, particularly when the feature map resolution increases. The higher the resolution, the larger the radius needed for cosine similarity computation, intensifying the computational challenge. In stark contrast, our method improves on this by computing similarities relative to clusters, not individual feature points, thus enhancing scalability.
> - **_"Trade-off between the memory footprint and performance speed:"_**
>
>   **Performance Comparison in Frames Per Second (FPS):**
>
>   | Resolution       | Feature Dimension | Ours  | INO  | DULVS |
>   |------------------|-------------------|-------|------|-------|
>   | 1/4 (120 x 210)  | 384               | 8.25  | 0.13 | 0.14  |
>   |                  | 768               | 6.50  | 0.11 | 0.12  |
>   |                  | 1152              | 5.25  | 0.10 | 0.11  |
>   | 1/2 (240 x 420)  | 384               | 1.55  | OOM  | OOM   |
>   |                  | 768               | 1.28  | OOM  | OOM   |
>   |                  | 1152              | 1.10  | OOM  | OOM   |
>
>   _OOM: Out of Memory_
>
>   We evaluated three models using feature maps at resolutions of 1/4 and 1/2 of the image size (480 x 840) across three feature dimensions (384, 768, 1152). This comparison involved computing the frames per second (FPS) after the first 21 frames, exclusive of the feature extraction step. Our model consistently outperformed the others, exhibiting particular strength in higher resolutions where competing models faced out-of-memory (OOM) issues.
>   (Tested using Tesla V100-32GB)

---

> ### Comment · Area_Chair_BjJk · 2023-08-20
> **Reminder from AC**
>
> Dear Reviewer
>
> Could you please check the rebuttal, if you have further concerns ?
>
> Best,
> AC

---

### Official Review · Reviewer_5uUQ · 2023-07-09

**Soundness:** 3 good
**Presentation:** 3 good
**Contribution:** 2 fair
**Rating:** 4
**Confidence:** 4

**Summary:**

The paper essentially shows, that a combination of classic techniques such as stream-data clustering, using an EM-algorithm and dynamic updates, in combination with strong pre-trained features, allows to achieve state-of-the-art performance on two standard video segmentation datasets (Davis 2017 and YouTube-VOS-2018).

**Strengths:**

Clearly achieving good performance with a combination of classic techniques, chosen well for the task at hand, is interesting to report and know about. Performance is strong on DAVIS (probably the dataset where most parameters are set) and somewhat less impressive on YouTube - but still sota

**Weaknesses:**

There is essentially no novelty in the paper - except to propose a well-chosen combination of classic techniques.

**Questions:**

The paper is well written and the performance is interesting to know about. I suppose the key question to me is essentially, how surprising the results are? If I had been asked prior to reading this paper if a sensible combination of stream-based clustering with strong pretrained features can do the job on datasets such as DAVIS and YouTube (these are not particularly difficult after all)  - I would have clearly set yes. In that sense the surprise and novel insight to me for this work is rather marginal. Having said this, I would assume that not everyone has the same intuitions and thus it is imho worthwhile to report the paper in some form as a publication. However, in my personal judgement the current paper and contribution seems below the bar for such a high-profile venue such as NeurIPS.

**Limitations:**

ok for me

---

> ### Author Rebuttal · Authors · 2023-08-09
>
> We thank the reviewer for the thoughtful and useful review.
> We also thank the reviewer for recognizing the strong performance of our method on both the DAVIS and YouTube datasets, as well as appreciating the integration of classic techniques.
>
> We would like to underscore the unique contribution that our work makes, which goes far beyond combining and adapting established techniques. The success of our method lies in the precision with which the techniques are adapted, tailored, and validated to deliver state-of-the-art results on benchmarks that continue to be challenging. The fact that after reading our paper the reviewer feels that the approach is logical and that it makes sense that it works, does not contradict the fact that no one did it before and does not dispute the novelty. Moreover, the details are crucial and many of our judicious choices are far from being obvious (e.g., opting to go with an ensemble of mixtures instead of model selection or nonparametric clustering techniques) and were always made while taking into account not only performance and principled modeling considerations but also efficiency and scalability.   In any case, we argue that if it were so obvious that setting new state-of-the-art results on a key computer-vision task can be done this way, people would have already done it before…
>
> Moreover, we are the first to demonstrate a low memory footprint in this domain, an essential quality in real-world applications. Our ability to easily scale to higher resolution features, while relying solely on static images during training, represents a breakthrough in unsupervised video object segmentation. This innovation not only builds on classic techniques but extends them in a way that could redefine best practices in the field.

---

> > ### Comment · Reviewer_5uUQ · 2023-08-18
> >
> > Thanks for the response. After reading all reviews and the rebuttals I essentially stand by my initial assessment. I can see that others are more positive and thus the paper will likely be accepted and I will not make the case to argue strongly against acceptance.
> >
> > As said, I find it personally below the bar for NeurIPS and that is still the case.

---

### Author Rebuttal · Authors · 2023-08-09

We thank the reviewers for their thoughtful reviews and constructive criticism. We are glad that, overall, the paper was positively received.

Here, in the general response, we touch upon several points common to more than one reviewer. Below, we reply to each reviewer separately.

### **Memory Utilization and Computational Load:**
The cumulative memory utilization during inference is affected by:
- **Memory Storage**: Traditional methods store multiple dense feature maps from prior frames, increasing the memory footprint based on the quantity and size of these maps. Our method significantly reduces this by tying memory usage to the number of clusters (as opposed to the much larger number of features), which demands far less memory than even a single dense feature map.
- **Computational Load**: Standard label propagation techniques compute cosine similarity for each feature point in the current frame with all feature points within a certain radius in all reference frames. That approach struggles with scalability, particularly when the feature map resolution increases. The higher the resolution, the larger the radius needed for cosine similarity computation, intensifying the computational challenge. In stark contrast, our method improves on this by computing similarities relative to the clusters present within a certain radius, not individual feature points, thus enhancing scalability (since the number of clusters within a region is much smaller than the number of features in that region).


### **Performance Comparison in Frames Per Second (FPS):**
| Resolution       | Feature Dimension | Ours  | INO  | DULVS |
|------------------|-------------------|-------|------|-------|
| 1/4 (120 x 210)  | 384               | 8.25  | 0.13 | 0.14  |
|                  | 768               | 6.50  | 0.11 | 0.12  |
|                  | 1152              | 5.25  | 0.10 | 0.11  |
| 1/2 (240 x 420)  | 384               | 1.55  | OOM  | OOM   |
|                  | 768               | 1.28  | OOM  | OOM   |
|                  | 1152              | 1.10  | OOM  | OOM   |
_OOM: Out of Memory_

We evaluated three models using feature maps at resolutions of 1/4 and 1/2 of the image size (480 x 840) across three dimensions (384, 768, 1152). This comparison involved computing the frames per second (FPS) after the first 21 frames, exclusive of the feature extraction step. Our model consistently outperformed the others, exhibiting particular strength in higher resolutions where competing models faced out-of-memory (OOM) issues.
(Tested using Tesla V100-32GB)


### **Adopting Deeper Networks:**
It is important to note that simply adopting deeper networks doesn't necessarily guarantee superior results. As observed in previous self-supervised learning approaches like VFS [1], performance sometimes remains unchanged or even degrades when using deeper backbones like ResNet-50. It can even introduce additional complications which already exist during training, such as issues with memory and convergence. Thus, our method provides an effective and much-needed solution to a practical problem.
The table below represents an evaluation conducted on the DAVIS-2017 validation set.
| Features                | J&F-Mean | J-Mean  | J-Recall | F-Mean  | F-Recall |
|-------------------------|----------|---------|----------|---------|----------|
| ours+dinov2_small       | 0.776    | 0.754    | 0.854    | 0.798    | 0.890    |
| ours+dinov2_base        | 0.782    | 0.760    | 0.880    | 0.804    | 0.900    |
| oursx2+dinov2_small     | 0.784    | 0.762    | 0.873    | 0.806    | 0.905    |
| oursx2+dinov2_base      | 0.803    | 0.780    | 0.885    | 0.825    | 0.916    |

In the experiment, DINOv2's [2] features were concatenated with our original features, a necessity due to DINOv2's large patch size of 14. This fusion provided more effective information utilization. The results showed a marked improvement in J&F-Mean with the "oursx2" configurations (features with higher spatial dimensions). This improvement is attributed to our method's ability to exploit and use higher resolutions, facilitated by XCiT's capability to handle them. A comparison between the DINOv2 versions ("small" vs "base") revealed that the "base" version consistently performed better, underscoring the influence of stronger pre-trained models. Overall, these findings demonstrate that the method's performance can be elevated by using higher resolutions and selecting more advanced pre-trained models, delineating a promising direction for future enhancements.

### **References:**
- [1] Xu, Jiarui, and Xiaolong Wang. "Rethinking self-supervised correspondence learning: A video frame-level similarity perspective." Proceedings of the IEEE/CVF International Conference on Computer Vision. 2021.
- [2] Oquab, Maxime, et al. "Dinov2: Learning robust visual features without supervision." arXiv preprint arXiv:2304.07193 (2023).

---

### Decision · Program_Chairs · 2023-09-21

**Decision:**

Accept (poster)

**Comment:**

Four reviewers vote for accept, and one reviewer gave borderline reject, and not willing to argue strongly against acceptance.

I would encourage the authors to take the reviewers' comments into consideration in the final camera ready paper.